# Monitoring the Spring 2021 Drought Event in Taiwan Using Multiple Satellite-Based Vegetation and Water Indices

Chien-Ben Chou [1,*], Min-Chuan Weng [1], Huei-Ping Huang [2], Yu-Cheng Chang [1], Ho-Chin Chang [1] and Tzu-Ying Yeh [1]

1    Meteorological Satellite Center, Central Weather Bureau, Taipei 10006, Taiwan
2    School for Engineering of Matter, Transport, and Energy, Arizona State University, Tempe, AZ 85281, USA
\*    Correspondence: joujb@cwb.gov.tw; Tel.: +886-2-2349-1311

**Abstract:** The monitoring of droughts is practically important yet challenging due to the complexity of the phenomena. The occurrence of drought involves changes in meteorological conditions, vegetation coverage and soil moisture. To advance the techniques for detecting and monitoring droughts, this study explores the usage of a suite of vegetation and water indices derived from high-resolution images produced by geostationary satellite Himawari-8. The technique is tested on the detection of the drought event in Spring 2021 across Taiwan due to deficit of precipitation in that season. It is found that the time series analysis of green chlorophyll index ($CI_{green}$) and normalized difference vegetation index (NDVI) helps detect the initiation of drought before its severity intensifies. The vegetation condition index (VCI) and vegetation health index (VHI) derived from $GI_{green}$ and NDVI are similarly useful for the early warning of a drought event. In addition to vegetation indices, the normalized difference water index (NDWI) is adopted for quantifying the deficit in precipitation. It is found that NDWI provides a better early warning system of drought compared to the vegetation indices. Combining the vegetation and water indices allows a more complete description of the evolution of drought for the Spring 2021 event. The potential for using the new framework for the early warning of future drought events is discussed.

**Keywords:** drought monitoring; Himawari-8; drought indexes





## 1. Introduction

Drought is a complicated phenomenon that involves multiple physical processes and spatiotemporal scales. Four types of droughts are classically defined [1]: Meteorological drought pertains to the condition with a deficit in precipitation; Agricultural drought develops when the shortage of water affects root-zone soil moisture and impacts agricultural production; Hydrological drought commences when the broader hydrological system—river, stream, reservoir levels and groundwater levels—are affected; Socioeconomic drought is declared when the aforementioned physical changes begin to affect human lives. Recently, a new type of flash drought has been considered [2,3]. It can intensify very rapidly from the onset of meteorological drought to fully developed agriculture drought, as the precipitation deficit is compounded by abnormally high temperatures, strong winds and an elevated level of sunshine all in the same area. A rapidly developing drought event can be particularly impactful to agriculture and human lives. Monitoring this type of event is of particular interest in this study.

The occurrences and societal impacts of flash droughts are major concerns in Tai-wan. Although the annual mean of precipitation in Taiwan is relatively high, around 2500 mm, a significant portion of the rainfall is concentrated in summer to fall, related to Mei-yu fronts and tropical cyclones. Rainfall in spring exhibits a greater degree of uncertainty. This fact, combined with steep topography over Taiwan which facilitates quick runoffs, makes spring a high-risk season for drought. Given that spring is the major plant growing season

in Taiwan, the impacts of springtime droughts on agriculture and human lives are often particularly severe. Due to complicated terrains and strongly heterogeneous vegetation coverage over Taiwan, monitoring early drought conditions is difficult when using ground-based observational methods. Thus, satellite remote sensing provides a potential alternative. To monitor rapidly developing droughts, measurements from geostationary satellites are particularly appealing due to their high spatiotemporal resolution.

Given this background, this study aims to develop techniques for sensing vegetation cover and hydrological properties to improve the early warning of droughts, using the newly available data from a geostationary satellite. This will be a pilot study to demonstrate the application of the geostationary satellite data. For this purpose, we apply the satellite-based indices to the study region over the island of Taiwan (*cf.* maps shown in Figures 1–5). The techniques will be tested by detecting and monitoring the evolution of the spring 2021 drought event in Taiwan.

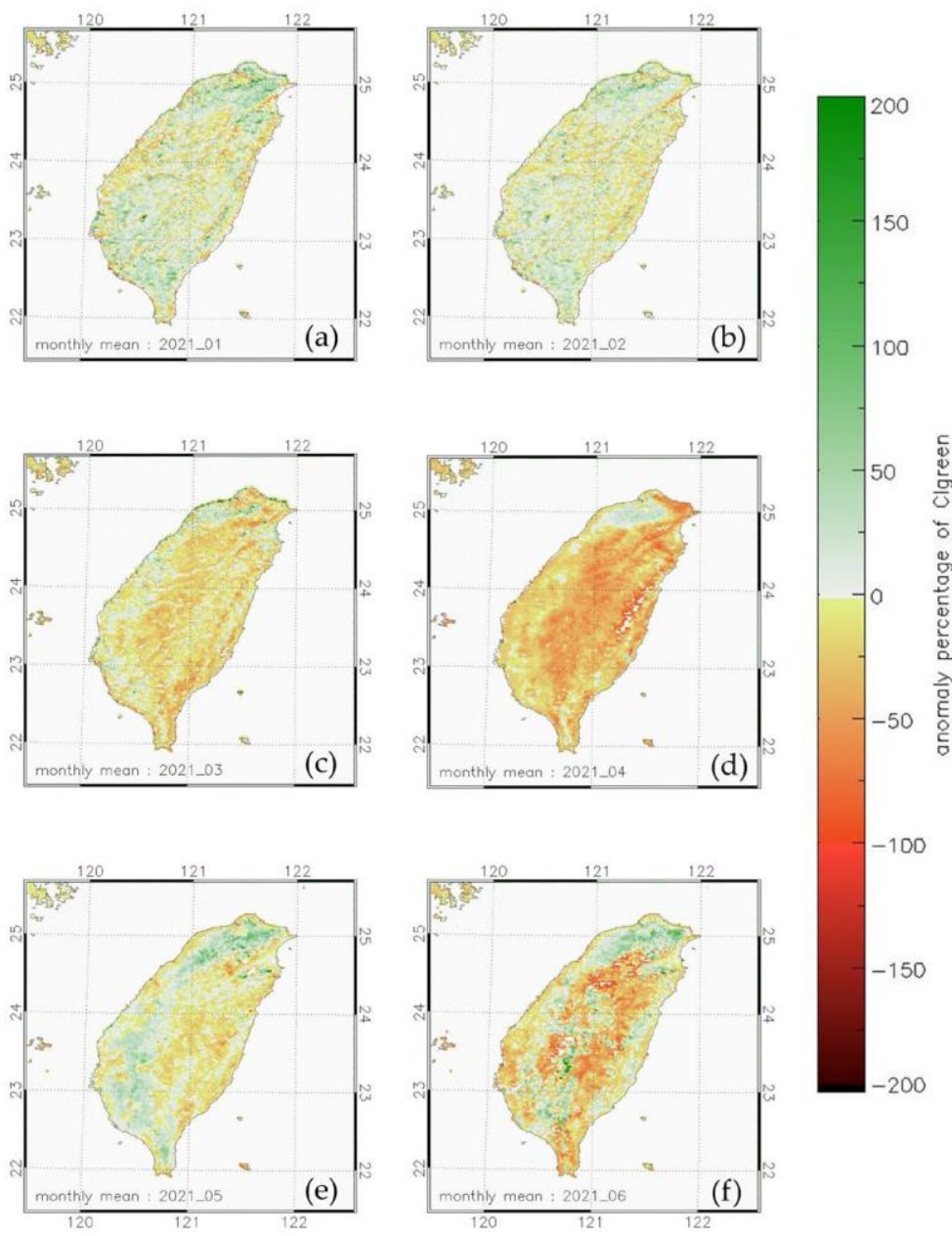

**Figure 1.** The anomaly of percentage of CI$_{green}$ over Taiwan from January to June 2021 (from (**a**–**f**)). Color scale is shown at the right. Latitude and longitude are labelled on the border.

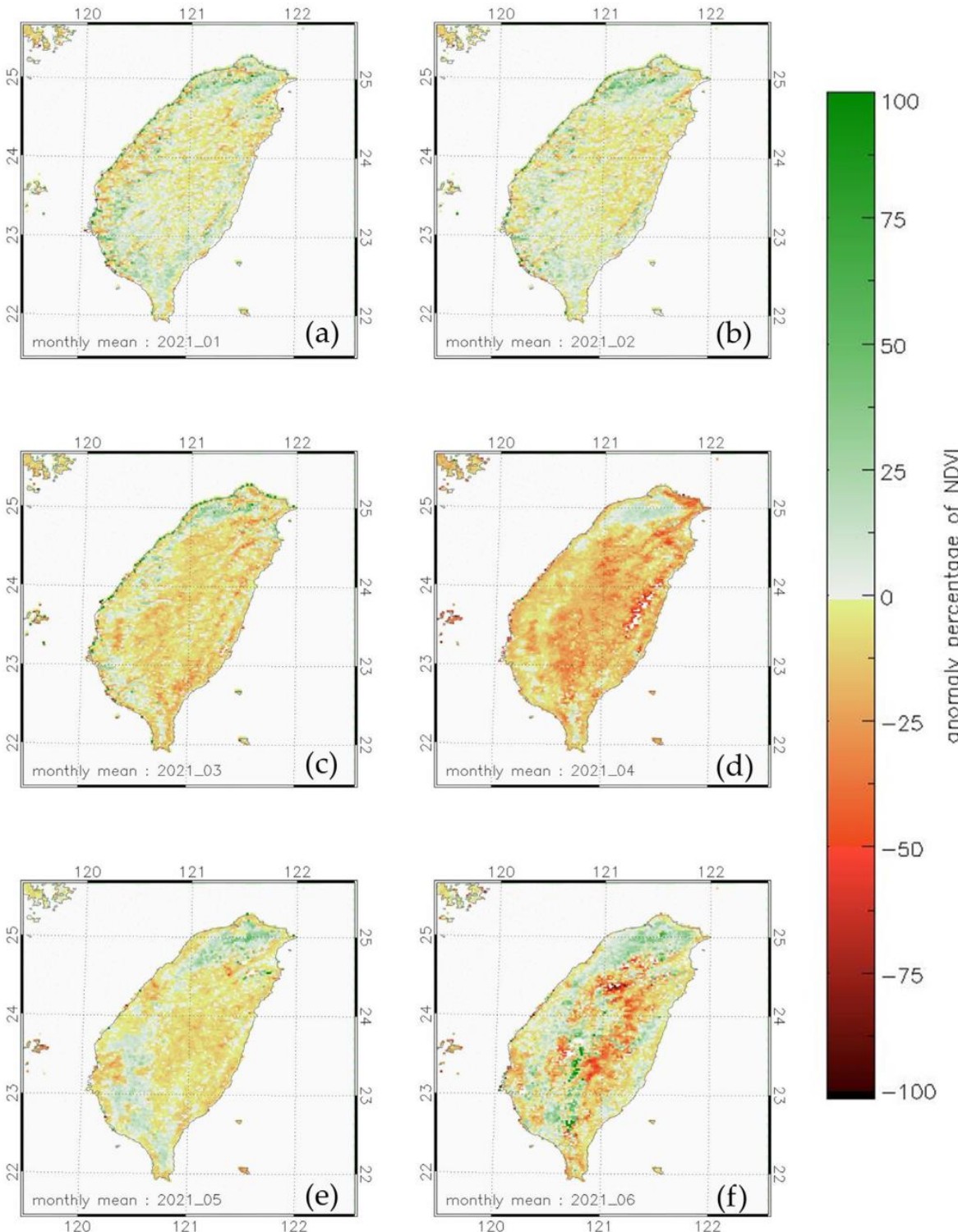

**Figure 2.** The anomaly of percentage of NDVI over Taiwan from January to June 2021 (from (**a**–**f**)). Color scale is shown at the right.

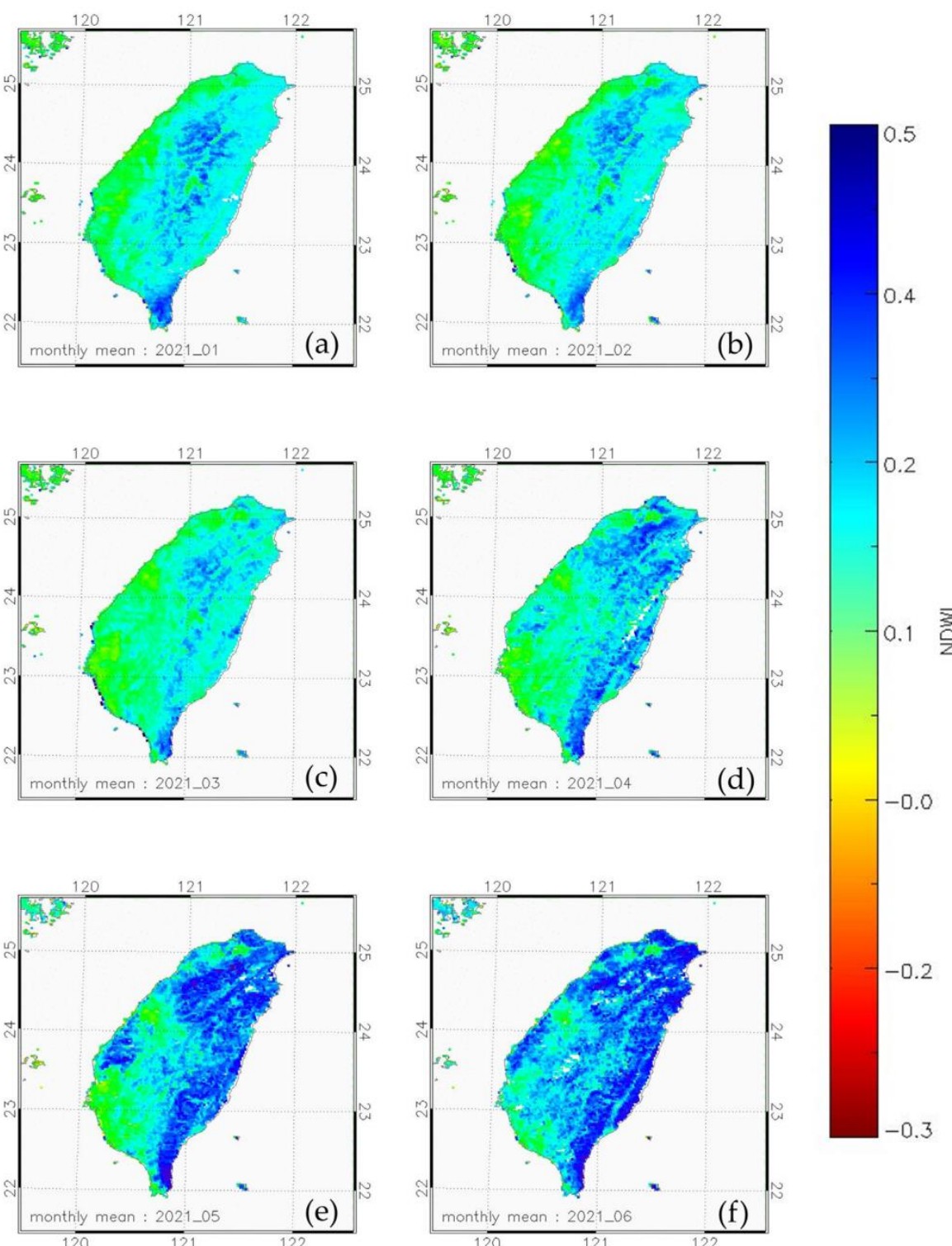

**Figure 3.** The mean value of NDWI (the version of [4]) over Taiwan from January to June 2021 (from (**a**–**f**)). Color scale is shown at the right.

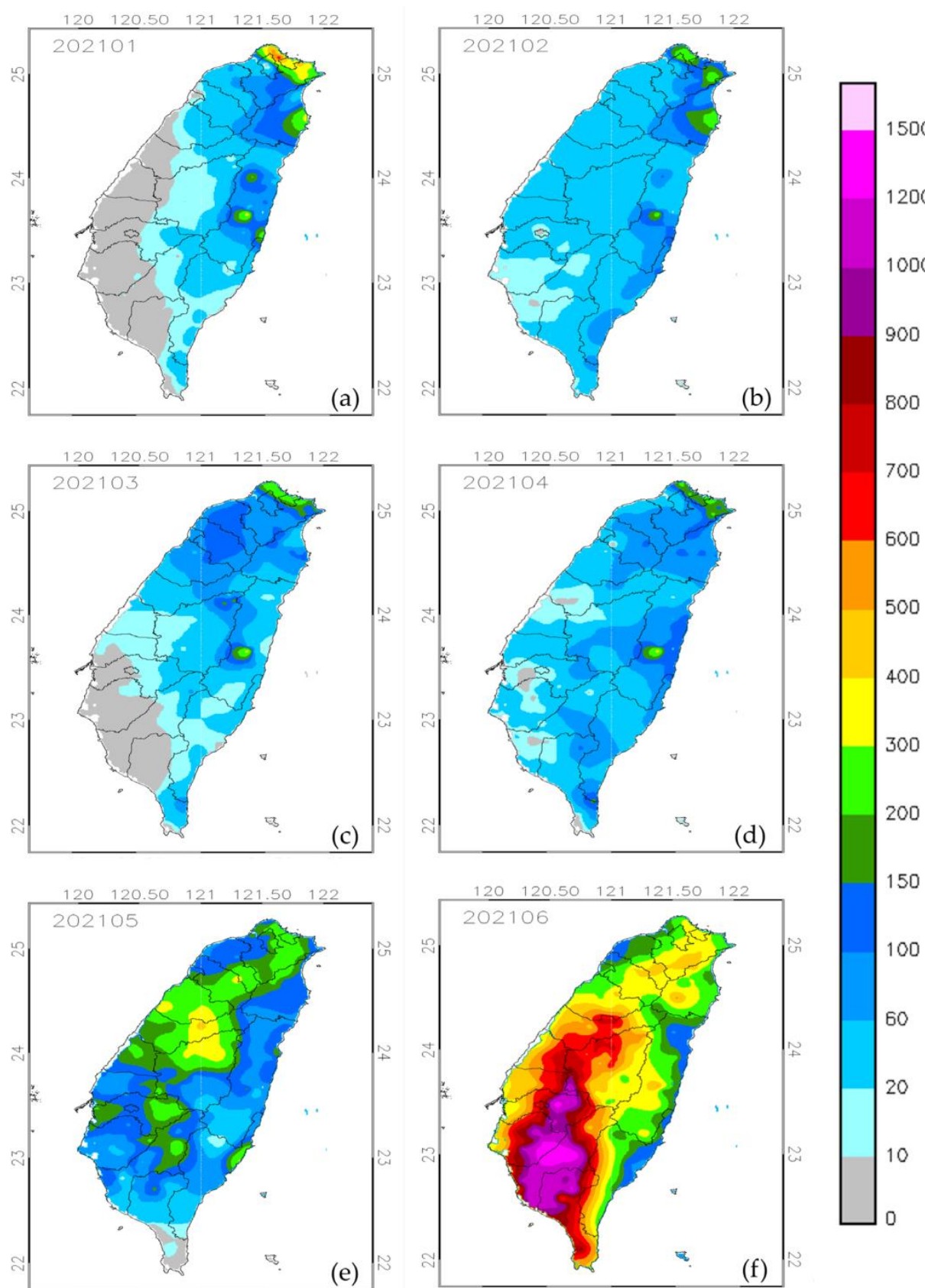

**Figure 4.** The monthly cumulative precipitation over Taiwan from January to June 2021(from (**a**–**f**)). The color scales are shown on the right.

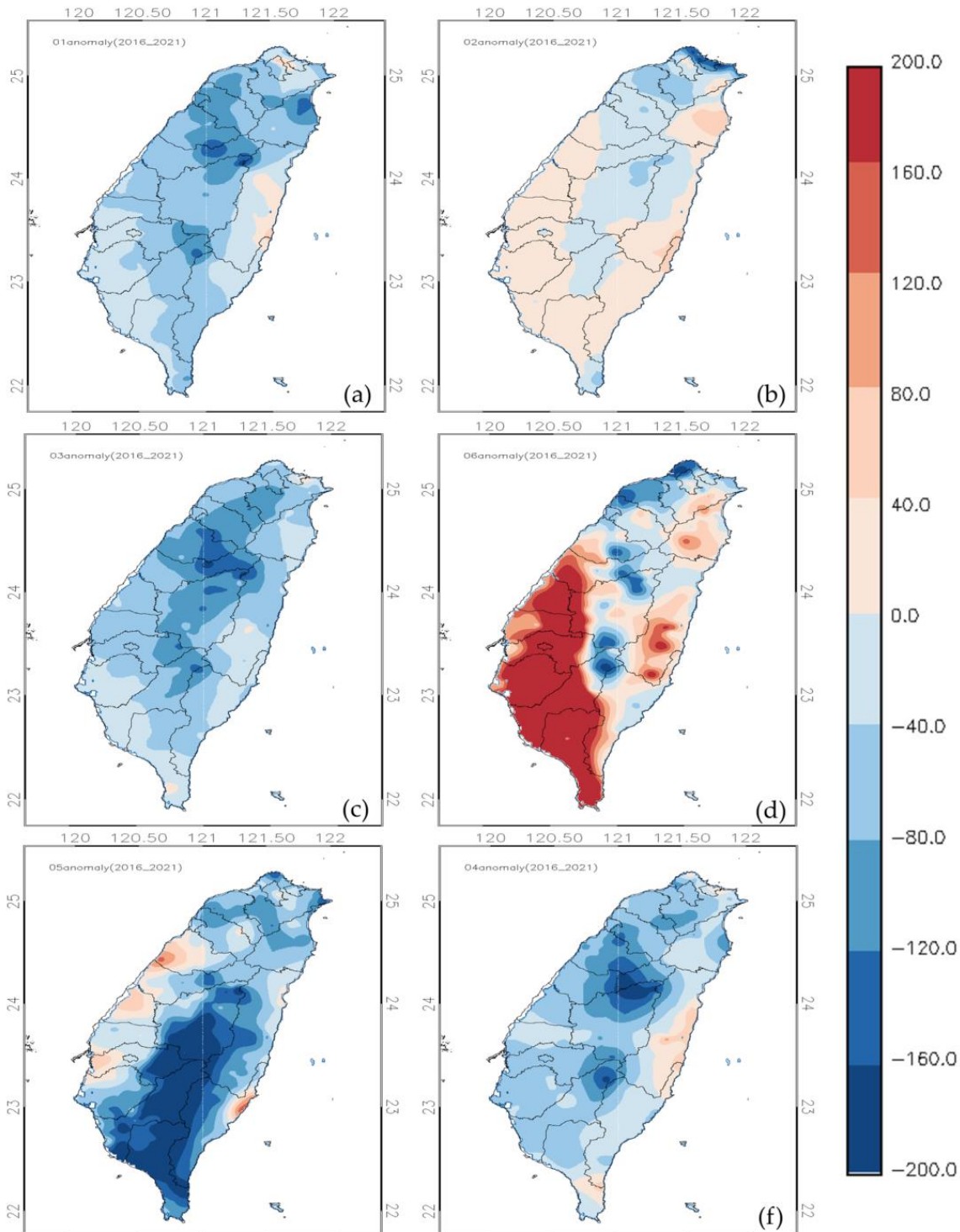

**Figure 5.** The anomaly in percentage of precipitation from January to June 2021(from (**a**–**f**)). The color scales are shown on the right.

　　　The key innovation of this work is in the application of geostationary satellite observation in a realistic setting of extreme drought. Conventionally, data from polar-orbiting satellites were more widely used for surveying land surface properties. With advances in instrumentation and improvement in algorithms, observation from a geostationary satellite has emerged as a desirable alternative. Our study focuses on detecting rapid changes in land cover using multi-channel geostationary satellite observation. This reflects a newer development of technology which complements the classical applications of geostationary

satellite data such as the monitoring of cloud maps over a broad area. With this background, our work will be based on the data from Himawari-8, an advanced geostationary satellite [5]. Further detail about the satellite is given in Section 2.

This study contributes to the latest research trend in implementing techniques that were previously used for a polar-orbiting satellite into their equivalents for a geostationary satellite, specifically for monitoring changes in land surface properties (e.g., [6]). With the increased number of observational channels on Himawari-8, such an implementation is now possible. In this context, the key parameters to be analyzed will include NDVI and green chlorophyll index ($CI_{green}$) [7], both are used to further derive the vegetation condition index (VCI) and vegetation health index (VHI) [8]. Given the time lag between the response in surface vegetation and the initial shortage of rainfall [9], we also include the normalized difference water index (NDWI) to detect the hydrological condition that complements the vegetation indices. While our focus is on the 2021 drought event, six years of AHI data from 2016 to 2021 are used in order to define the anomalies for that particular event. Thus, we establish a framework of drought detection using the observation from a geostationary satellite alone. The potential future applications of the technique for monitoring flash droughts are discussed.

## 2. Data and Methodology

### 2.1. Study Area and Geostationary Satellite Observation

The study domain was the island of Taiwan, approximately bounded by 120–122° E and 22–25.5° N (the domain is shown through Figures 1–5). The key qualities to analyze concerning droughts are vegetation and water coverage at the surface. Due to the lack of high-resolution in situ observation network over the complex terrains of the island, the analysis relies exclusively on satellite remote sensing. Key indices for vegetation and water will be deduced from satellite measurements, as detailed in the following section.

Himawari-8 is a new-generation geostationary meteorological satellite launched by the Japan Meteorological Agency on 7 October 2014, which became fully operational on 7 July 2015. The satellite is located at 140.70E over the equator. It covers most of East Asia and the Western Pacific Ocean, and its counterpart in the southern hemisphere. Being geostationary, the instrument onboard the satellite has a high-temporal resolution of 10 min, superior to what is available with polar-orbiting satellites. The key instrument, Advanced Himawari Imager (AHI), has 16 channels, 3 which are visible, 3 which are near-infrared and 10 which are infrared [5], which is a significant upgrade from its predecessor, the geostationary Multifunction Transport Satellite-2 (MTSAT-2). In addition to the visible (red) channels at 0.63 µm already on MTSAT-2, the AHI is equipped with two visible (blue and green) channels at 0.47 and 0.51 µm, with a resolution of 1 km, 1km and 0.5 km, respectively, and three new near-infrared channels at 0.86, 1.6 and 2.3 µm with a resolution of 1 km, 2 km and 2 km, respectively. The availability of the new channels on AHI allows a more versatile construction of the vegetation and water indices. The ten infrared channels on AHI also have improved spatial resolution at 2 km, compared to 4 km for its predecessor. The two channels at 10.4 and 12.4 µm can be used to estimate the land surface temperature [10].

As a tradeoff of using the data from the newest satellite, the available record from Himawari-8 is relatively short. Our analysis is based on 6 years of observation data, collected since the launch of the satellite. Keeping this in mind, this work focuses on the proof of concept in establishing a technical framework for drought detection. We anticipate that the technique can be used more widely in the future when longer observations by geostationary satellite become available.

### 2.2. Overview of Vegetation and Water Indices

To account for multiple physical processes associated with droughts, we analyze a set of indices for vegetation and hydrological conditions, derived from measurements by AHI. The usage of similar indices for monitoring land cover has a long history [11]. For satellite remote sensing, classical indices such as normalized difference vegetation index

(NDVI) and its variations are well known (e.g., [12,13]). While following similar ideas for the construction of those indices, we customize the definitions for them based on available observations from the geostationary satellite.

The application of vegetation and water indices draws inspiration from previous studies (mostly based on polar-orbiting satellite observation), which we now briefly review. Vegetation indices for drought monitoring have long been used (e.g., [11]), starting from the well-known NDVI which uses near-infrared and red reflectance to measure the concentration of chlorophyll in the canopy. Variations in the index have since been considered. For example, enhanced vegetation index (EVI) enhances the ability to detect higher levels of biomass ([12,13]). Several indices are defined to incorporate the effect of soil moisture, for example soil-adjusted vegetation index (SAVI) [14], modified soil-adjusted vegetation index (MSAVI) [15] and optimized soil-adjusted vegetation index (OSAVI) [16]. Amid the proliferation of those definitions, a universal normalized vegetation index (UNVI) has also been recently proposed [17]. The study suggests that UNVI, NDVI and EVI have a similar nonlinear behavior in response to chlorophyll content. Namely, saturation occurs in UNVI, NDVI and EVI when the chlorophyll content reaches a critical level. Related works have been carried out to define alternative indices, notably $CI_{green}$ and its variants, which behave linearly and are not subject to saturation at a high concentration of chlorophyll ([18,19]). Thus, in terms of nonlinear vs. linear behavior, there are two groups of indices—NDVI and its variants and $GI_{green}$ and its variants—which serve as the candidates for our study. While the definitions of the vegetation indices vary among existing studies, they follow similar principles of extracting the surface properties from the contrast between different channels. In this work, we follow the same principles but define the relevant indices with necessary adjustments (detailed below), taking into account the available channels in AHI on the Himawari-8 satellite.

To prepare for the analysis of the drought-detecting indices, the data from AHI observations are first processed by the Clavr-x package to generate the cloud mask product. This helps classify the data pixels into clear, probable clear, probable cloudy and cloudy conditions. Only the data identified under a clear-sky condition are selected for the computation of the vegetation and water indices. We process the data from 2016 to 2021, using the mean from this period to contrast the anomalies in 2021.

For the vegetation indices, we first generate NDVI and $CI_{green}$, then use them to derive VCI and VHI. The NDVI is used because of its long history of success in applications for detecting vegetation growth. As surveyed above, $CI_{green}$ is chosen for its linear relationship to the chlorophyll concentration in vegetation. This will complement NDVI which is more prone to saturation at a high concentration. For this purpose, the high concentration of chlorophyll can indeed be frequently encountered over Taiwan, due to the subtropical climate and vegetation types in the region. The AHI on Himawari-8 has enough channels of observation to allow the construction of both NDVI and $CI_{green}$. Together with VCI and VHI, the set of four indices are used for monitoring vegetation. A water index, NDWI, is also computed to aid in the detection of drought. The details for the indices are given in the following.

*2.3. Normalized Difference Water Index (NDWI)*

Following [20], the normalized difference water index (NDWI) is computed by

$$\text{NDWI} = \frac{\rho_{GREEN} - \rho_{NIR}}{\rho_{GREEN} + \rho_{NIR}} \tag{1}$$

where $\rho_{GREEN}$ is the reflectance at visible green spectral band and $\rho_{NIR}$ is the reflectance at near-infrared band. The Himawari-8 data on channel 2 with central wavelength of 0.51 μm and channel 4 with central wavelength of 0.86 μm are chosen to represent $\rho_{GREEN}$ and $\rho_{NIR}$, respectively. The value of NDWI ranges from $-1$ to 1. A higher positive value means a higher ratio of coverage of wet surfaces over the observational area. The definition in [20] sets zero as a threshold for interpreting that the coverage by wet surfaces is

everywhere. Since the threshold could vary with data resolution and surface characteristics, other studies have considered a non-zero threshold. For example, it is set to 0.1 in [21] to account for severe flooding events and set to 0.3 in [22] for other general applications.

Another widely adopted version of NDWI was considered by [4]. This index is sensitive to changes in liquid water content in the vegetation canopies. It is defined by

$$\text{NDWI} = \frac{\rho_{NIR} - \rho_{SWIR}}{\rho_{NIR} + \rho_{SWIR}} \tag{2}$$

where $\rho_{NIR}$ is the reflectance at near infrared band with the central wavelength of 0.86 μm and $\rho_{SWIR}$ is the reflectance at short-wave infrared band with the central wavelength of 1.24 μm. We used Himawari-8 data on channel 4 with central wavelength of 0.86 μm to represent $\rho_{NIR}$. Since Himawari-8 measurement does not include a channel at 1.24 μm, we used a slightly modified definition by using the nearby channel 5 with central wavelength of 1.61 μm to represent $\rho_{SWIR}$.

### 2.4. Normalized Difference Vegetation Index (NDVI)

The normalized difference vegetation index (NDVI) has been extensively used for monitoring vegetation dynamics from space [11]. It is computed by

$$\text{NDVI} = \frac{\rho_{NIR} - \rho_{RED}}{\rho_{NIR} + \rho_{RED}} \tag{3}$$

where $\rho_{NIR}$ and $\rho_{RED}$ are the reflectance in near-infrared band and visible red band. The Himawari-8 data on channel 4 with central wavelength of 0.86 μm and channel 3 with central wavelength of 0.64 μm are chosen to represent $\rho_{NIR}$ and $\rho_{RED}$, respectively. For detecting the vegetation condition over land, the value of NDVI ranges from 0 to 1. It is close to zero over rock or bare soil. Higher values NDVI represent more extensive coverage of vegetation under less stressed conditions.

### 2.5. Green Chlorophyll Index (CI$_{green}$)

The green chlorophyll index (CI$_{green}$) can be used to estimate the total chlorophyll content that indicates the physiological status of vegetation [18]. A highly linear relationship is found between CI$_{green}$ and canopy chlorophyll content, which makes the former an accurate estimator of the latter [23]. The index is computed by

$$\text{CI}_{\text{green}} = \frac{\rho_{NIR}}{\rho_{GREEN}} - 1 \tag{4}$$

where $\rho_{NIR}$ and $\rho_{GREEN}$ are the reflectance at near-infrared band and visible green band. Similar to the treatment of the other indices, the Himawari-8 channel 2 (0.51 μm) and channel 4 (0.86 μm) data are used to represent $\rho_{GREEN}$ and $\rho_{NIR}$, respectively.

### 2.6. Vegetation Condition Index (VCI)

The vegetation condition index (VCI), formulated by [24], represents the value of NDVI for a specific observation normalized by the difference between maximum and minimum values of NDVI from long-term observations. It is computed by

$$\text{VCI}_{\text{m}} = 100 \left( \frac{NDVI_m - NDVI_{min}}{NDVI_{max} - NDVI_{min}} \right) \tag{5}$$

where $NDVI_m$ is the averaged value of NDVI over a focused period (which will be one month for our later applications) and $NDVI_{min}$ and $NDVI_{max}$ are the minimum and maximum values of NDVI from long-term observations (which will be from 2016 to 2021). We affix the subscript "m" to VCI in Equation (5) as a reminder that the index characterizes the vegetation condition over the particular month of our focus. The VCI indicates the level of water-related stress on the plants, assuming that other non-water related factors

are in normal conditions (for example, no damages to the plants by pests). The value of VCI ranges from 0 to 100. A higher value indicates a lower level of water-related stress endured by the plants.

*2.7. Temperature Condition Index (TCI)*

The temperature condition index (TCI) is analogous to the VCI but for quantifying the stress level for vegetation related to land surface temperature [25]. It is computed by

$$TCI_m = 100 \left( \frac{LST_{max} - LST_m}{LST_{max} - LST_{min}} \right) \tag{6}$$

where $LST_m$ is the land surface temperature (LST) averaged over a focused period (one month) and $LST_{min}$ and $LST_{max}$ refer to the minimum and maximum values of LST from long-term observations (2016 to 2021). Here, land surface temperature is readily available as a direct output of the Clavr-x pre-processing package and does not require additional calculations. With its value ranging from 0 to 100, a higher value of TCI indicates that the plants are under less thermal stress. Following [24,26], the LST in Equation (6) can be replaced by brightness temperature from satellite observations.

*2.8. Vegetation Health Index (VHI)*

The vegetation health index (VHI) is a linear combination of VCI and TCI, computed by

$$\text{VHI}_m = \alpha \left( VCI_m \right) + (1 - \alpha)(TCI_m) \tag{7}$$

where $\alpha$ is a constant between 0 and 1. As a weighted average of $VCI_m$ and $TCI_m$, this index combines the influence of wetness and thermal stresses on the plants. The most appropriate value of $\alpha$ to take depends on the type of vegetation and other background physical conditions. As this aspect has not been extensively investigated, for this study we choose $\alpha = 0.5$ based on [27,28].

## 3. Results

After the suite of vegetation and water indices are computed, we apply the derived data to the monitoring of the spring 2021 drought event in Taiwan. This event commenced in the beginning of 2021 and reached its most severe phase in April that year. A moderate amount of rainfall at the end of April and the beginning of May provided relief to the drought condition. The evolution of the event is illustrated in Figures 1 and 2 by the maps of the anomaly of percentage of CI$_{green}$ and NDVI from January to June 2021. Hereafter, the anomaly is defined with respect to the 2016–2021 mean. The brown color indicates a negative anomaly. It can be seen that a negative anomaly of vegetation coverage occurred over the whole Taiwan area, with a weak but noticeable anomaly already occurring in January. The decrease in vegetation indices accelerates in March, with the minimum reached in April. As a check of robustness, the two sequences of maps based on CI$_{green}$ and NDVI are consistent with each other. As a reference to the mean hydrological condition of this period, the bottom row of Figure 3 shows the maps of the NDWI based on [4] as computed from Equation (2). Up until April, the baseline level of wetness is moderate. Then, it picks up significantly in May and June.

To supplement the description of drought based on vegetation indices, we show the evolution of precipitation over spring 2021. Figure 4 is the monthly cumulative rainfall from January to June. Figure 5 shows the corresponding percentage anomalies. The peak drought condition in April can be seen as the response to the deficit in precipitation since the beginning of 2021 (particularly in January, March and April). Interestingly, while there was a negative anomaly in precipitation (particularly over the mountains) in May, the drought condition eased in that month. This reflects a crucial point we mentioned in the Introduction: In spring, with an already low climatological rainfall, a deficit in precipitation can greatly suppress plant growth. In contrast, in summer, when the climatological precipitation is

abundant, a moderate deficit does not damage the vegetation. In fact, in May 2021, the total amount of rainfall in that month was enough to aid some recovery in vegetation. In June 2021, the precipitation anomaly finally turned positive with the arrival of Mei-yu fronts. From Figures 1 and 2, in June, some clusters of positive vegetation anomaly began to emerge. Nevertheless, there are also spots where vegetation indices become more negative compared to May. This is not due to a new drought condition, but rather possibly damages to plants caused by excessive rainfall and stormy conditions associated with the Mei-yu systems. Again, this can also be inferred from the maps of NDWI in the bottom row of Figure 3.

To assess the strategy for monitoring the drought, we next analyze the time evolution of the vegetation and water indices averaged over the island of Taiwan. To provide the contrast between the spring season and the rest of the annual cycle, in Figures 6–8 we show the area-averaged indices stepping through 12 months of the year. The blue line and accompanying pair of gray lines represent the mean and plus/minus one standard deviation of the monthly mean value of an index as derived from the data for 2016–2021. The black line shows the corresponding value for 2021 alone. In this format, Figure 6a shows the 12-month evolution of the green chlorophyll index $CI_{green}$. The climatological value of this index reaches the minimum in March and maximum in July, then staying at the plateau through the rest of the year. For the 2021 event, in March and April, the value of $CI_{green}$ dips significantly below the already low climatological mean for the respective months. This could be adopted as a criterion to declare a region-wide drought condition. In terms of the tendency in time, the decreasing trend is the steepest from February to March. Such a steep trend provides an early warning for the severe drought that is yet to come in April.

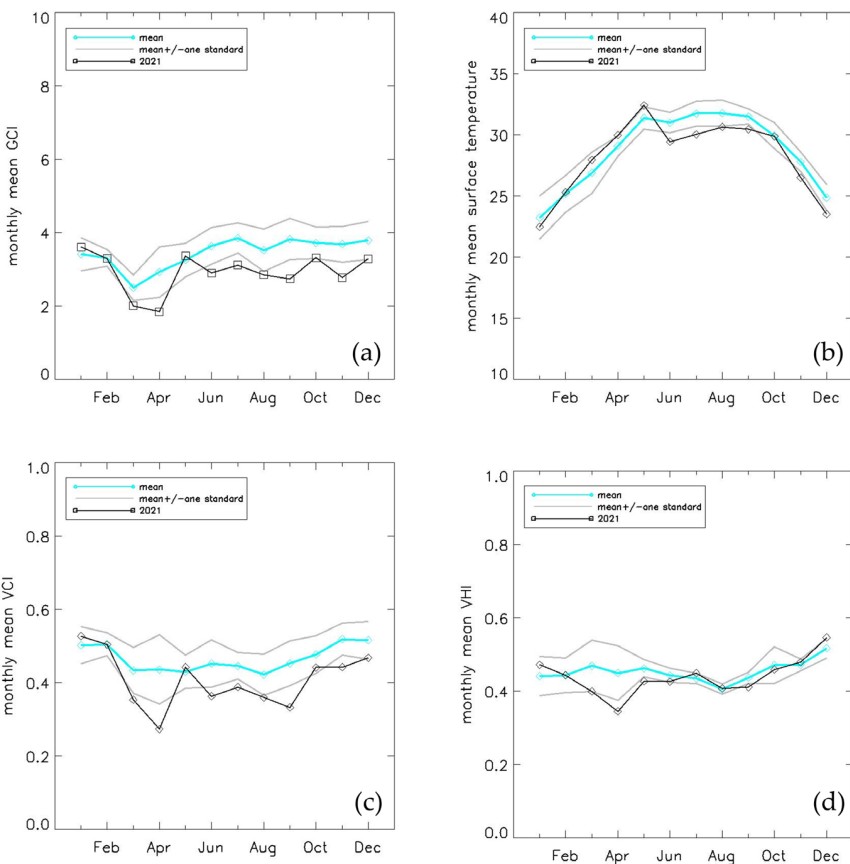

**Figure 6.** The time series of monthly mean indices and variables stepping through the 12-month annual cycle (January to December, as marked on the abscissa). For each index, the blue line and the accompanying pair of gray lines are the mean and +/− one standard deviation derived from the data for 2016–2021. The black line is the value of the index for 2021. (**a**) $CI_{green}$; (**b**) surface temperature; (**c**) VCI; (**d**) VHI.

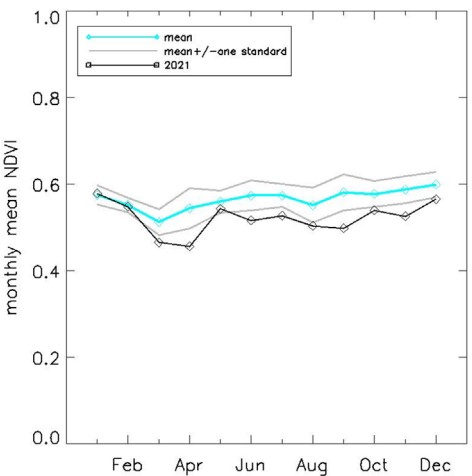

**Figure 7.** The time series of NDVI presented in the same format as Figure 6.

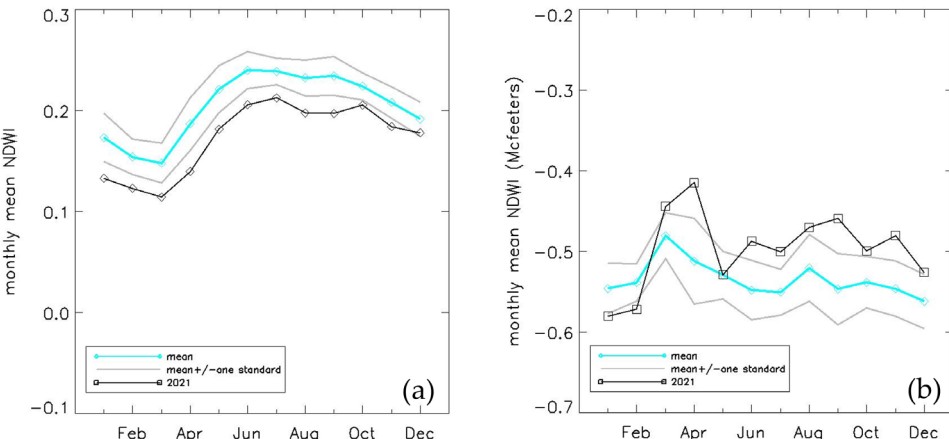

**Figure 8.** The time series of (**a**) the [4] version of NDWI and (**b**) the [20] version of NDWI. Presented in the same format as Figure 6.

In Figure 6b–d, we further show the 12-month evolution of (b) surface temperature, (c) vegetation condition index (VCI) and (d) the vegetation health index (VHI). In spring 2021, surface temperature was generally above normal which was a favorable condition for drought. The evolution of VCI in Figure 6c is similar to that of CI$_{green}$ in Figure 6a, as the former is derived from the latter. The VHI shown in Figure 6d combines the indices of surface temperature (TCI in Equation (6)) and vegetation (VCI). It also shows a negative anomaly through spring in 2021. Note that the range of plus/minus one standard deviation for this index is relatively large in spring, implying a higher risk for droughts to occur in this season. In contrast, the standard deviation is small in summer, implying a less volatile condition for the occurrence of droughts in the warm season. Since VCI and VHI depend on CI$_{green}$, which is in turn computed from NDVI, we show the area-averaged time series of NDVI Figure 7. With the occurrence of a significantly negative anomaly in spring 2021, the behavior of NDVI is broadly similar to VCI or VHI. All three indices are potentially useful for monitoring drought. The particular choice might depend on applications. (For example, VHI includes the information on surface temperature and might be a sharper indicator for a hypothetical drought that is caused by severely elevated temperature).

While the vegetation indices represent an important aspect of the drought, particularly its agricultural impacts, for the purpose of early warning, it is known that negative anomalies in water indices emerge even before the decline in vegetation indices [9]. To supplement the vegetation indices, we further show the time evolution of NDWI in Figure 8, using the same format as Figures 6 and 7. Figure 8a is for the version of NDWI defined in [4]. It shows a negative anomaly (at one standard deviation below normal) in wetness starting

from January 2021. In this context, this index potentially provides a sharper indication of early warning of drought compared to the vegetation indices. On the other hand, after January–February, the negative anomaly in NDWI stays at about the same level through the rest of the year. It does not show a steep decline in March as do the vegetation indices. Thus, the vegetation indices provide a better representation of the rapid intensification of drought leading to its peak in April. The insight here is that different indices reflect different aspects of the drought. We may obtain a more complete picture of a drought event by combining a hybrid set of those indices together.

We caution that a wetness index such as NDWI only quantifies the fractional coverage of wet surfaces. It might not always reflect the amount of water (soil moisture) through the depth of soil that is crucial for plant growth. As such, the decoupling between the evolution of NDWI and vegetation indices is not surprising. The interpretation of NDWI is further complicated by the fact that different versions of the index do not always behave in a similar manner. For example, Figure 8b shows the evolution of NDWI defined by [20] instead of [4] (see Equations (1) and (2) for definitions). This index is designed such that it is negative for vegetation cover, zero for barren soil and positive for water [20]. In Figure 8b, a significant increase in the index that occurs in March–April reflects the dominant change in land cover from vegetation to barren soil as drought intensifies. Thus, for this event, this version of NDWI tells us more about vegetation than water, and Figure 8b looks rather similar to Figure 6a or Figure 7 flipped upside down.

This study relies exclusively on satellite remote sensing products. Ideally, validations with in situ measurements would further strengthen the results. At the high resolution over the heterogeneous landscape as illustrated in Figures 1–3, a comparable ground-based observational network to cross-validate satellite data does not yet exist. It is expected that such validations could be carried out in the future, perhaps through a dedicated field experiment.

The preceding results demonstrate a framework of using a geostationary satellite for monitoring drought. Thus, we achieve the goal of transforming relevant techniques from the setting of a polar-orbiting satellite to a geostationary satellite. This new research theme has emerged as the new generation of geostationary satellites, such as Himawari-8, became available less than a decade ago. Given the relatively short record of data and few extreme drought events within the observational period, what we have accomplished by studying the 2021 event is a "proof of concept". If longer observations including more extreme drought events become available from Himawari-8 and its successors, future studies can extend our framework to create a statistical model for not only monitoring but also predicting drought. We envision that such a model will use the suite of parameters analyzed in this paper, and their time derivatives, as the main input.

Lastly, we only analyzed monthly means despite the potentially higher resolution in time for geostationary satellite. The relatively conservative choice is influenced by the shortness of available record. With longer observations, future studies could consider analyzing changes in surface conditions based on weekly or pentad means, which will be particularly useful for monitoring rapidly developing droughts. We also note that while the shortness of the data and smallness of samples for the extreme events do not yet allow an extensive statistical analysis, such an analysis should be carried out in the future with longer data. As an example of what kind of analysis can be carried out, let us consider a statistical test for the significance of the anomaly of the vegetation index in Figure 6a (or Figures 6c and 7). It drops outside one standard deviation for two consecutive months in March and April. Given the typical time scale of synoptic weather system of about 1 week, and quick runoff (due to steep topography) to flush away the influence of a rainfall event on vegetation, one might nominally take 1 week as one independent degree of freedom. This would give 8 degrees of freedom over two months, enough to establish an over 95% level of significance for a shift in the mean of the index (See, e.g., Equation (1) in [29] for the quick estimate for *t*-test; the criterion for 95% significance being $N \geq 8 \, (\Delta/\sigma)^{-2}$, where $N$ is degree of freedom, $\Delta$ is the shift in the mean and $\sigma$ is one standard deviation. Thus,

with $\Delta/\sigma \approx 1$, we need $N \geq 8$.). However, a more rigorous estimate of degrees of freedom should be carried out through an analysis of the correlation in time (and possibly space) using sufficiently long observations. It is in this context that longer data will be needed. We look forward to pursuing this direction in the future.

### 4. Concluding Remarks

This study demonstrates the technique of using a suite of vegetation and water indices to monitor droughts. Those indices are derived from measurements by the geostationary satellite Himawari-8. The indices incorporate information on vegetation coverage, surface temperature and surface wetness. They are used to reconstruct the evolution of a major drought event in Taiwan in spring 2021. It is found that the vegetation indices exhibit a sharp decline from February to March that year, corresponding to the rapid development of the drought towards its peak in April. Moderate negative anomalies in the vegetation indices are present in January–February. The water index NDWI shows complementary behaviors: A more significant negative anomaly in the index already appeared in January, making it a sharper indicator for the early warning of drought. However, in February–March, the water index does not exhibit a steep decline similar to the vegetation indices. Surface temperature is also found to be above normal in spring 2021, which plays a secondary role in exacerbating the drought condition. The key lesson learned is that different indices reflect different physical aspects of the drought and combining them together leads to a more complete picture of a drought event for its monitoring and detection.

This study has focused on one major drought event in spring 2021. Through the case study, the indices derived from geostationary satellite observations are proven useful for monitoring the evolution of drought. With this encouraging result, we recommend further applications of the technique to future drought events. With the accumulation of longer observations from a geostationary satellite, we may be able to build a statistical model that uses satellite-based vegetation and water indices to perform seasonal prediction on droughts.

**Author Contributions:** Data curation, H.-C.C. and T.-Y.Y.; Investigation, C.-B.C.; Methodology, C.-B.C. and M.-C.W.; Writing—review & editing, H.-P.H. and Y.-C.C. All authors have read and agreed to the published version of the manuscript.

**Funding:** This research received no external funding.

**Institutional Review Board Statement:** Not applicable.

**Informed Consent Statement:** Not applicable.

**Data Availability Statement:** Not applicable.

**Acknowledgments:** The authors appreciate the comments from reviewers through the revisions of this paper.

**Conflicts of Interest:** The authors declare no conflict of interest.

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
