# Peer review of "Monitoring the Spring 2021 Drought Event in Taiwan Using Multiple Satellite-Based Vegetation and Water Indices"

_atmosphere, doi:10.3390/atmos13091374_

Round 1

Reviewer 1 Report (Previous Reviewer 2)

Thanks for your revision, there are some minor revison should be done as follows: 1)Namely, saturation occurs in NUVI, NDVI and EVI when the chlorophyll content reaches,line 130, NUVI should be UNVI?Please check it.

Author Response

Reviewer 2 Report (Previous Reviewer 3)

Dear Authors,

My comments have been included in the PDF file. Please follow up in your revision of the manuscript.

Good luck.

Round 2

Reviewer 2 Report (Previous Reviewer 3)

Dear Authors,

Thank you for your efforts in this work and the revised version of your manuscript.

All the best.

This manuscript is a resubmission of an earlier submission. The following is a list of the peer review reports and author responses from that submission.

Round 1

Reviewer 1 Report

The current version requires more work. With six co-authors, I think you can deliver more solid research. At its current stage, it sounds like it has potential, but it is hard from my perspective to see the novelty of the work. I like the methodology and the dataset used, but it did not deliver a complete research article. 

Reviewer 2 Report

The author studied the drought events using Hima-8 data using NDVI,NDWI and other parameters, however, there some points need to be evaluated before it can be accept.

the main problems is lack of backgroud explaination and new references should be added.

please see the attached document.

Reviewer 3 Report

Dear Authors,

Authors must totally rewrite the manuscript in accordance with the rules of academic writing.

2. The authors should organize their paper manuscript as follows: abstract, introduction, study area, materials and methodology, results, discussion, and conclusions, followed by a list of sources.

3. Authors must submit professionally-designed, high-resolution figures, maps with descriptive titles that indicate the figures' contents.

4. Enriching the introduction with a greater examination of recently published sources and expanding the number of citations in the list of references.

5. Analyze the results using statistical analysis professionally.

5. In order to monitor a specific event, such as drought, it is necessary to observe the occurrence over an extended period of time, such as twenty or thirty years, and compare it to the average for the research area.

Good luck.